# Commensal and Pathogenic Members of the Dental Calculus Microbiome of Badia Pozzeveri Individuals from the 11th to 19th Centuries

**DOI:** 10.3390/genes10040299

**Published:** 2019-04-12

**Authors:** Tasha M. Santiago-Rodriguez, Antonio Fornaciari, Gino Fornaciari, Stefania Luciani, Isolina Marota, Giuseppe Vercellotti, Gary A. Toranzos, Valentina Giuffra, Raul J. Cano

**Affiliations:** 1Diversigen Inc., Houston, TX 77021, USA; 2Department of Translational Research on New Technologies in Medicine and Surgery, Division of Paleopathology, University of Pisa, 56128 Pisa, Italy; v.giuffra@med.unipi.it; 3Department of Civilizations and Forms of Knowledge, University of Pisa, 56128 Pisa, Italy; ginofornaciari@gmail.com; 4Laboratory of Molecular Archaeo-Anthropology/ancient DNA, School of Biosciences and Veterinary Medicine, University of Camerino, 62032 Camerino, Italy; stefania.luciani@unicam.it (S.L.); isolina.marota@unicam.it (I.M.); 5Department of Anthropology, The Ohio State University, Columbus, OH 31901, USA; vercellotti.2@osu.edu; 6Department of Biology, University of Puerto Rico, San Juan, PR 00931, USA; gary.toranzos@upr.edu; 7The BioCollective, Denver, CO 80014, USA; raul.cano@thebiocollective.com

**Keywords:** ancient microbiomes, dental calculus, oral microbiome, respiratory pathogens

## Abstract

The concept of the human oral microbiome was applied to understand health and disease, lifestyles, and dietary habits throughout part of human history. In the present study, we augment the understanding of ancient oral microbiomes by characterizing human dental calculus samples recovered from the ancient Abbey of Badia Pozzeveri (central Italy), with differences in socioeconomic status, time period, burial type, and sex. Samples dating from the Middle Ages (11th century) to the Industrial Revolution era (19th century) were characterized using high-throughput sequencing of the 16S ribosomal RNA (rRNA) gene V4 region. Consistent with previous studies, individuals from Badia Pozzeveri possessed commensal oral bacteria that resembled modern oral microbiomes. These results suggest that members of the oral microbiome are ubiquitous despite differences in geographical regions, time period, sex, and socioeconomic status. The presence of fecal bacteria could be in agreement with poor hygiene practices, consistent with the time period. Respiratory tract, nosocomial, and other rare pathogens detected in the dental calculus samples are intriguing and could suggest subject-specific comorbidities that could be reflected in the oral microbiome.

## 1. Introduction

The oral microbiome is the second most characterized after the gut microbiome, and it is known to possess over 700 bacterial species belonging to the Firmicutes, Proteobacteria, Actinobacteria, and Bacteroidetes [1]. Approximately 57% of the known oral species are named, 13% remain unnamed but are cultivable, and 30% still remain uncultivated [2]. Characterization of the human oral microbiome also led to the realization that it can differ according to geographical regions and culture [3]. For instance, the oral microbiome of isolated Amerindians is distinct and more diverse compared to subjects with westernized lifestyles [4]. Other studies characterized the oral microbiome in association with diverse disease phenotypes including dental caries and periodontitis [5,6,7]. The human oral microbiome is also known to differ and form complex interactions according to biogeographical sites, which include the tongue, palate, and saliva, as well as subgingival and supragingival plaque [8,9]. Of these oral biogeographical sites, dental plaque harbors bacteria that could reflect oral health and disease, lifestyles, and dietary habits [10]; yet, a limited number of studies addressed the association of disease, dietary habits, and lifestyles with the oral microbiome of westernized and isolated modern human societies using dental plaque. 

Until recently, the concept of the human oral microbiome was applied to better understand health and disease, lifestyles, and dietary habits through human history [11,12,13]. Similarly to modern human societies, ancient oral microbiomes from Europe and the Caribbean are also dominated by bacteria from the main phyla including the Firmicutes, Bacteroidetes, Actinobacteria, and Proteobacteria, and seem to be more diverse than westernized oral microbiomes [11,12,13]; however, studies of ancient human oral microbiomes are still limited to certain European and Caribbean groups, and more recently the Neanderthals [11,12,13,14]. In the present study, we augment the understanding of the ancient oral microbiome by characterizing dental calculus of human remains dating from the Middle Ages (11th century) to the Industrial Revolution era (19th century) in chronological order using high-throughput sequencing of the 16S ribosomal RNA (rRNA) gene V4 region. The human dental calculus samples were recovered from the archaeological site of San Pietro di Pozzeveri, in the village of Badia Pozzeveri located at about 10 km southeast of the city of Lucca (Tuscany, Italy). The ongoing excavation is aiming to understand socioeconomic status, lifestyles, dietary habits, and disease of the people from this geographical region. While the dental calculus samples are from the same archeological site, these are diverse in terms of time period, socioeconomic status, burial method, and gender, providing a unique opportunity to elucidate potential changes in the oral microbiome as a result of the mentioned factors (Table 1).

## 2. Materials and Methods

### 2.1. History of Excavation Site and Sample Description

The study focuses on the archaeological site of “San Pietro di Pozzeveri”, in the village of Badia Pozzeveri, located about 10 km southeast of the city of Lucca (Tuscany, Italy) (Figure 1). The ongoing archaeological excavation of the site was performed by the University of Pisa (Division of Paleopathology) and by the Ohio State University (Department of Anthropology) since 2011 in an attempt to reconstruct the architectural evolution of the monastery and to obtain a large set of osteological samples for bioarcheological studies of the Medieval and modern Tuscan population [15]. The archeological excavation was conducted by the “open area method”. Briefly, five large areas, covering a total surface of about 750 m^2^ around the church of San Pietro (Figure 1), were opened during the 2011–2017 archaeological campaigns to explore the presence of burials and to understand the developmental plan of the Medieval monastic buildings.

Several religious buildings were established over the centuries within the area. For instance, the Abbey of Pozzeveri, with the church of San Pietro, was founded at the end of the 11th century on the site of a Medieval settlement formed by a church (documented in the year 1039), as well as a rectory founded in the year 1056 [15]. The Abbey of the Camaldolese Order flourished during the 12th and 13th centuries as a result of its location along the Via Francigena, which was a major trade and pilgrimage route connecting France and northern Europe with Rome throughout the Middle Ages. The decline of the monastery started in the 14th century and led to its dissolution at the beginning of the 15th century. The church of San Pietro survived as a religious center of the scattered settlements of the region from the 16th to 20th centuries and assumed the role of a parish church with an annexed cemetery. The funerary use of the space around the church covers a period from the 11th to 19th century. The older graves, in simple ground trenches, are related to the first church (11th century). Burials of the time of the Abbey (12–13th centuries) vary in terms of typology and spatial location. A group of individuals were buried in simple ground trenches in the space of the parvis in front of the façade. The excavation also revealed within the same location four lithic coffins, indicating privileged burials, and each contained a single skeletal individual. Several lithic coffins, with many skeletal individuals inside, were built against the north side of the nave and against the façade of the church in the 13th century. These are collective privileged burials for prominent lay families who obtained the special privilege to be buried in monumental structures placed against the church. Privileged grave spaces for the monks were observed in the ambulatory of the cloister (12–13th centuries). In the Modern Age, a churchyard was delimited by a wall in front of the parish church with burial in simple ground tranches. The first phase of the Modern cemetery was during the 16–17th centuries, and the second phase was during the 18th century. The old parish churchyard was abandoned at the beginning of the 19th century, and a new cemetery was built where burials in simple ground trenches with wooden coffins can be found. A specific sector possesses burials in ground trenches with the bodies covered by layers of lime related to the cholera epidemics of 1855. The anthropological age of individuals was established by the degree of dental wear, the morphology of pubic symphysis, and the costosternal junction of ribs [16,17,18]. The sex was determined by the pelvic and cranial morphology [19].

### 2.2. Sample Collection, DNA Extraction, and Avoidance of Contamination

Dental calculus was sampled from individuals of the different burial phases of Badia Pozzeveri, covering a chronology from the Middle Ages (11th century) to the Industrial Revolution Era (19th century) (Table 1). Sampling took place immediately after digging of the skeletons, in a protected environment (i.e., vertical laminar flow safety hood), using sterile gloves, a mask, and surgical gowns. The sampled individuals did not show dental abscesses, inflammatory signs of alveolitis, or periodontitis. The dental calculus samples were collected following the current suggested protocols to avoid cross-contamination [20,21]. Briefly, the isolated tooth, skull, or jaw was placed over clean aluminum foil to collect fragments of the calculus, which consisted of supragingival plaque from molars and pre-molars. Dental calculus was removed from the lingual or buccal surface of the teeth with a sterile scalpel, used only once for each sample to minimize the risk of cross-contamination, and then collected in individual sterile 15-mL centrifuge polypropylene tubes. The tubes were kept in a dry, darkened environment at 18–20 °C until their shipping to the Laboratory of Molecular Archaeo-Anthropology/Ancient DNA of the University of Camerino, Italy. This laboratory is exclusive for ancient DNA work, with no modern samples handled in this facility. All dental calculus samples were handled following standard precautions for ancient DNA work, as extensively described previously [20,21,22]. Safety cabinets, dedicated gel trays, tanks, and reagents were all ultraviolet (UV)-radiated before and after every use. 

To avoid contamination, all DNA extractions were conducted in the laboratory of Molecular Archaeo-Anthropology/Ancient DNA in the University of Camerino, Italy. An initial decontamination step with UV irradiation was applied to the dental calculus samples for 1 min, followed by pulverization and washing in 0.5 M ethylenediaminetetraacetic acid (EDTA) for 15 min with agitation. Subsequently, the dental calculus samples were digested for 24 h at 55 °C with agitation in a lysis buffer containing 0.5 M EDTA (pH = 8), SDS 2%, and proteinase K (20 mg/mL). DNA was then isolated using QIAamp DNA Investigator kit following manufacturer’s instructions (Qiagen, USA). DNA from soil samples (250 mg) was extracted using the MoBio PowerSoil DNA Isolation kit (Carlsbad, CA, USA), and two samples were further included in the sequencing pool. Soil samples were processed separately from the dental calculus samples to avoid cross-contamination. While an extraction blank was included for every two to three samples, one extraction blank was included in the sequencing run. An extraction blank from the QIAamp DNA investigator kit was also included in the sequencing run, as recommended previously, to assess the extent of contamination due to the reagents [23]. 

### 2.3. 16S rRNA Gene High-Throughput Sequencing

Sequencing of the 16S rRNA gene V4 region was performed at Molecular Research Laboratory (MRDNA) (www.mrdnalab.com; Shallowater, TX, USA). DNA samples were handled in areas exclusively used for PCR amplification and sterilized using DNAaway and UV radiation. Template manipulations were handled in separate hoods that were also sterilized using DNAaway and UV radiation. Non-template PCR controls were included, as recommended previously, in the PCR and showed no amplification or a DNA signal; thus, they were not included in the sequencing run [23]. The 16S rRNA gene V4 variable region was amplified using the PCR primers 515 forward (F) (GTGCCAGCMGCCGCGGTAA) and 806 reverse ® (GGACTACHVGGGTWTCTAAT) using a single-step, 30-cycle PCR using the HotStarTaq Plus Master Mix Kit (Qiagen, USA) as follows: an initial denaturation at 94 °C for 3 min, followed by 28 cycles of denaturation at 94 °C for 30 s, annealing at 53 °C for 40 s, and elongation at 72 °C for 1 min, followed by a final elongation step at 72 °C for 5 min. The V4 region was selected because it is well characterized compared to other regions, and the amplified product is within the length considered to be appropriate in ancient DNA analyses (<300 bp). PCR products were then checked in a 2% agarose gel, mixed in equal concentrations, and purified using Agencourt AMPure beads (Agencourt Bioscience Corporation, MA, USA). Library preparation was conducted using the Illumina MiSeq reagent kit V3 (2 × 300 bp) following the manufacturer’s instructions.

### 2.4. 16S rRNA Gene Sequence Analyses 

Paired-end fastq files were joined using Quantitative Insights into Microbial Ecology (QIIME) using join_paired_ends.py. Barcodes were removed using split_libraries.py with default filtering parameters (http://qiime.org/scripts/split_libraries.html). For comparison, sequences from four healthy saliva (accession numbers SRS017345, SRS013228, SRS013185, SRS064596), four healthy subgingival (accession numbers SRS022119, SRS014323, SRS023328, SRS021816), and four healthy supragingival plaque (accession numbers SRS013252, SRS042272, SRS020893, SRS022315) samples, as well as four subgingival plaque samples (accession numbers SRR3503166, SRR3503259, SRR3503292 and SRR3503326) from individuals with periodontal disease, were included in the analyses [24]. Additional information about these samples is listed in Appendix A. Operational taxonomic units (OTUs) were selected using the pick_closed_reference_otus.py workflow. Then, 16S rRNA taxonomy was defined by ≥97% similarity to the default database. Before further diversity and taxonomic analyses, Bayesian microbial source tracking was performed using SourceTracker to identify possible sources of contamination, as well as the relative proportions of contaminant sources, including both soil samples from the archeological site and the extraction blank control [25]. Sources included the four saliva, four subgingival, and four supragingival plaque samples from healthy individuals, as well as four subgingival plaque samples from individuals with periodontitis, two soil samples from the archeological site, and one extraction blank. Once the relative proportions of contaminant sources originating from the soil samples and the extraction blank were determined, OTUs present in the soil samples from the archeological site and the extraction blank control were removed from all the samples using the script workflow described in http://qiime.org/tutorials/filtering_contamination_otus.html before any further analyses. The OTU tables corresponding to the soil and extraction blank control samples were then merged with the filtered oral samples using the merge_otu_table.py script.

Data were rarefied to 2388 sequences to minimize the effect of disparate sequence number on the results. Then, α- and β-diversity, and taxonomy assignments were determined using the core_diversity_analyses.py workflow based on sample type (i.e., dental calculus, plaque with periodontitis, saliva, subgingival plaque, supragingival plaque, soil, and blank control), century (i.e., 11th–19th), social status (low/middle, good, noble/high, and high-status Camaldolese monk), burial type (lithic coffin and simple burial trench), and gender (i.e., male and female). Furthermore, α-diversity metrics of the bacterial communities, including observed OTUs and Shannon diversity, were compared using the compare_alpha_diversity.py script using default parameters. Then, β-diversity (Bray–Curtis) was visualized as a principal coordinate analyses (PCoA) plot using the principal coordinate file as input for the make_2d_plot.py script. The shared_phylotype.py script was run to determine the number of shared OTUs between dental calculus and modern oral microbiome samples. Results from the shared_phylotype.py matrix were then visualized as a heatmap constructed using the function heatmap.2 available in the R package gplots. Group significance analyses were also run using the group_significance.py with default parameters to determine any patterns in specific OTUs across modern and ancient oral microbiomes, as well as associations with the other factors including century, social status, burial type, and gender.

## 3. Results

Twenty-six different dental calculus samples from the archaeological site of “San Pietro di Pozzeveri” were characterized using 16S rRNA gene high-throughput sequencing of the V4 region (Table 1). Prior to further analyses, SourceTracker analyses were performed to identify potential sources of environmental contamination including soil from the archeological site, and contamination due to laboratory reagents (extraction blank control). The proportion of OTUs in the samples resembling soil contamination and reagent contamination are shown in Figure 2A. OTUs (genus level) present in the extraction blank (Dataset 1) and soil samples (Datasets 2 and 3) were then removed from the ancient and modern oral microbiome samples to resolve the identification of potential authentic oral bacteria (Figure 2B). The number of OTUs prior- and post-filtering are shown in Appendix A. Furthermore, β-diversity was visualized in a PCoA plot to determine the effect of filtering the OTUs identified in the blank control and soil samples from the dental calculus samples. Results show that the dental calculus samples and extraction blank control clustered separately, and this separation was significantly different (*p*-value = 0.035; *R*^2^ = 0.086; *F*-statistic = 2.34). This suggests that the OTUs present in the extraction blank control did not significantly affect the interpretation of the results (Appendix A). In addition, β-diversity of the dental calculus samples prior to and after filtering the extraction blank and soil samples OTUs showed that these clustered closely to the extraction blank and soil samples. After filtering the OTUs identified in the extraction blank and soil samples, the dental calculus samples clustered separately (*p*-value = 0.001; *R*^2^ = 0.296; *F*-statistic = 6.88) (Appendix A). When plotting the percentage of OTUs prior to and after filtering, as well as those identified in the extraction blank control and soil samples, results show that soil may have had a greater number of identified OTUs impacting the dental calculus samples (Appendix A; Dataset 4). The proportion of OTUs in the dental calculus microbiomes that are shared with modern oral microbiomes was determined (Figure 2C). While all the dental calculus samples shared OTUs with the modern oral microbiome samples, 19 dental calculus samples had a higher proportion of OTUs and clustered separately from the modern oral microbiome samples (Dataset 5; Figure 2C). This was further demonstrated with the higher number of observed OTUs (Figure 3A) in the dental calculus samples after filtering the OTUs identified in the soil samples from the archeological site, as well as those present in the extraction blank control. Observed OTUs were also determined for the dental calculus samples based on century (Figure 3B), social status (Figure 3C), burial type (Figure 3D), and gender (Figure 3E). Shannon diversity was also determined based on sample type (Figure 3F), century (Figure 3G), social status (Figure 3H), burial type (Figure 3I), and gender (Figure 3J).

Furthermore, β-diversity based on the Bray–Curtis distances was visualized as PCoA plots. Analyses were performed to visualize differences based on sample (Figure 4A), century (Figure 4B), social status (Figure 4C), burial type (Figure 4D), and gender (Figure 4E). There was a separation of the data based on sample type, and this separation was significantly different (*p*-value = 0.001; *R*^2^ = 0.252; *F*-statistic = 2.14) (Figure 4A). Expectedly, the modern and ancient oral microbiomes clustered into separate groups, while the soil samples and extraction blank controls clustered closely (Figure 4A). Data were also plotted to visualize differences based on century, showing a statistically significant separation (*p*-value = 0.01; *R*^2^ = 0.307; *F*-statistic = 1.14) (Figure 4B). No statistically significant separation of the β-diversity distances was noted based on social status (Figure 4C), burial type (Figure 4D), or gender (Figure 4E).

Taxonomic composition of the dental calculus samples after filtering the OTUs identified in the soil samples from the archeological site and extraction blank control showed that the most abundant phyla were the Acidobacteria, Actinobacteria, Bacteroidetes, Firmicutes, and Proteobacteria. The relative abundances of each of the most abundant phyla based on sample type, including dental calculus, dental plaque of subjects with periodontitis, saliva, subgingival, and supragingival plaque from healthy subjects, soil samples from the archeological site, and the extraction blank control, are shown in Figure 5A. The relative abundances of the included phyla based on century are presented in Figure 5B. Due to the limited number of samples for certain groups, no significant comparisons could be made at the phylum level based on social status; nevertheless, the high-status Camaldolese monk had lower Bacteroidetes and Firmicutes relative abundances, but higher Actinobacteria relative abundances compared to the other social groups (Figure 5C). The relative abundances of the five main phyla based on burial type (Figure 5D) and gender were similar, and no significant differences were noted (Figure 5E).

Further analyses focused on oral and nasopharynx bacterial species to aid in the interpretation of the results. The average relative abundances of selected bacterial species identified in both ancient and modern samples are shown in Table 2. Other bacterial species seemed to be unique to the dental calculus samples (Dataset 6; Table 2). Interestingly, bacterial species with different cellular metabolisms were identified only in the most recent samples (mid-19th century) including aerobes (*Neisseria bacilliformis*), facultative anaerobes (*Enterococcus dispar* and *Actinobacillus succinogenes*) [26,27] and strict anaerobes (*Bifidobacterium infantis*, *Selenomonas noxia*, and *Veillonela parvula*) (Dataset 7) [28,29]. Oil-associated bacteria including *Methylibium petroleiphilum* [30], bacteria isolated from fermented vegetables including *Lactobacillus harbinensis* [31], *Ignatzschineria indica* (a rare human pathogen) [32], and *Desulfosporomusa polytropa* (a soil-associated bacterium) (Dataset 8) [33] were only identified in the high-status Camaldolese monk. No significant differences were noted in the mean relative abundances of bacterial species based on burial type (Dataset 9) or sex (Dataset 10) after adjusting the *p*-values.

## 4. Discussion

### 4.1. Environmental Microbiome of Badia Pozzeveri Dental Calculus Remains

Characterization of ancient human oral microbiomes in association with health and disease, lifestyles, and dietary habits is now possible due to high-throughput sequencing technologies; however, data interpretation is usually limited due to post-mortem contamination [20,24]. Thus, it is essential to include soil samples from the archeological site(s), as well as an extraction blank control, to determine the extent of the contamination of the sample(s) of interest. The present study included the mentioned samples, as well as a standard non-template PCR reaction as controls [23]. A number of bioinformatic approaches are used to authenticate ancient oral microbiomes when using 16S rRNA gene high-throughput sequencing, which can include SourceTracker analyses [11,13,25]. SourceTracker analysis enables the identification of possible external or contaminant OTUs, and their subsequent filtration or removal from an OTU table, enabling the resolution of the authentic ancient oral microbiome of interest. A number of OTUs in the dental calculus samples included in the present study matched OTUs present in the soil samples from the archeological site, as well as the extraction blank control. Soil-associated bacterial OTUs were still identified in the dental calculus samples even after removing or filtering bacterial OTUs present in the soil samples from the archeological site. This may suggest that soil-associated bacteria were able to colonize the dental calculus samples during exposure to the burial site or could be a result of accidental geophagia. It is also possible that the soil-associated bacteria in the external environment changed throughout the centuries, while the soil-associated bacteria that were able to colonize the dental calculus samples remained relatively unaltered; thus, these soil-associated bacteria that were able to colonize the dental calculus samples may possibly reflect some of the bacterial communities of the soil samples at the time of deposition. Shotgun metagenomic sequencing may have possibly enabled the identification of additional species that may have been associated with post-mortem contamination; however, given that 16S rRNA gene databases have a higher number of sequences that can be classified at various taxonomical levels compared to the number of complete genomes available in databases, 16S rRNA gene high-throughput sequencing may have provided more information regarding the extent of the post-mortem contamination compared to shotgun metagenomic sequencing [34]. Nevertheless, data are still informative in terms of identifying oral-specific bacterial OTUs in the dental calculus samples, which seem to include both commensals and pathogens.

### 4.2. Commensal and Pathogenic Microbiome in Badia Pozzeveri Remains

The present study focused on the oral and nasopharynx microbiome. Consistent with previous studies applying 16S rRNA gene high-throughput sequencing of dental calculus and dental pulp samples, individuals from Badia Pozzeveri from the 11th to 19th centuries possessed commensal oral bacteria that resembled those known to inhabit modern oral microbiomes [11,12,13,14,35]. These results suggest that members of the oral microbiome are ubiquitous in terms of presence, despite differences in geographical regions, time period, sex, and socioeconomic status (although these factors can also alter the relative abundances); yet, differences in the taxa that were detected only in the dental calculus samples could not be explained entirely by century, social status, burial type, or sex, possibly due to the limited number of samples per group. In addition, it is important to consider that oral hygiene practices and hygiene practices in general were limited, which may explain an extent of these results, such as the presence of fecal bacteria including *Enterococcus cecorum* and *Enterococcus faecium*.

Interestingly, identification of certain bacterial species in the ancient samples could also suggest that some of the subjects suffered from periodontitis. While the presence of periodontitis-associated pathogens, including members of the red complex (i.e., *Tannerella forsythia*, *Treponema denticola*, and *Porphyromonas gingivalis*), was previously examined in dental calculus samples, sequences from the 16S rRNA gene V4 region of these bacteria were not detected in the samples included in the present study [13]. This could possibly be due to the limitations of the 16S rRNA gene V4 region for species-level resolution; however, the presence of *Solobacterium moorei* and *Desulfomicrobium orale* in the dental calculus and dental plaque samples of subjects with periodontitis, but not in the samples of the healthy subjects, soil, or the extraction blank controls could suggest that certain individuals from Badia Pozzeveri may have suffered from inflammatory conditions. For instance, *S. moorei* was identified as part of the oral microbiome of individuals with oral lichen planus [36]; yet, data should be carefully interpreted due to the limitations of the 16S rRNA gene V4 region for species-level resolution and that the presence of certain bacteria may not always indicate disease. In addition, the sampled individuals did not show dental abscesses, inflammatory signs of alveolitis, or periodontitis.

The presence of opportunistic pathogens associated with cystic fibrosis, such as *Burkholderia gladioli* and *Inquilinus limosus* [37,38], respiratory tract infection bacteria such as *Tsukamurella pulmonis* [39], potential nosocomial pathogens including *Methylobacterium mesophilicum* [40], *Microbacterium oxydans*, which was isolated from human throat swabs [41], bacteria isolated from peritoneal carcinomatosis, intra-abdominal infections, and scleromas including *Lentzea albidocapillata*, *Necropsobacter rosorum*, and *Arthrobacter scleromae*, respectively [42,43,44], is intriguing and could suggest that these subjects could have been affected by respiratory illnesses and other opportunistic pathogens. However, data should be carefully interpreted due to limitations of sequencing one 16S rRNA gene variable region for species-level resolution. Moreover, the presence of the DNA of these potentially opportunistic pathogens may not necessarily indicate disease. For instance, some microorganisms, as in the case of *Staphylococcus aureus*, can colonize the oral tract of a small percentage of the healthy population at a specific time point (intermittent colonization), and may not always result in disease progression [45]. Other described bacteria may also inhabit other environments outside the human oral cavity and nasopharynx.

### 4.3. Dietary Habits from Badia Pozzeveri and Oral Microbiome

The present study also attempted to identify bacteria that could be associated with specific lifestyles and dietary habits. The reason is that much of what is known about medieval lifestyles comes from historical accounts, mostly pertaining the sociopolitical and religious elite; thus, information about the lower classes is less readily available. While archeological and isotope information indicates that medieval food access followed strict demarcations based on age, sex, and status, the dental calculus microbiome did not reflect these differences. This could possibly be due to a lack of preservation of certain bacterial DNA in the more ancient samples, or the low number of samples included per group. Isotopic studies confirmed the presence of animal origin products in the diets of rural sires between the 11th and 14th centuries, but people had mostly a vegetarian diet. C4 cereals such as broomcorn millet, foxtail millet and sorghum were consumed from the 11th–14th centuries and were considered low-quality. C3 crops, on the other hand, were consumed by the upper class because they are low in carbohydrate content. Interestingly, isotope data support similar diets between status groups and gender in children, with increasing disparities including limited access to animal protein into adulthood, especially for males. These specific dietary habits could be potentially shown with some of the environmental taxa identified; however, the interpretations could be limited due to the risk of environmental contamination. Interestingly, *L. harbinensis*, which was originally isolated from fermented vegetables and was also identified in fermented drinks, was only identified in the high-status Camaldolese monk [31]. This could suggest that the high-status Camaldolese monk diet included exclusive items that were not part of the middle/low-class diet.

## 5. Conclusions

While ancient microbiome studies can be challenged by external contamination, particularly that coming from soil, sequence data obtained from calculus samples give us a unique glimpse of the oral microbiomes of ancient cultures. From the data, it is evident that ancient oral microbiomes can include both commensal and pathogenic microorganisms, many of which may reflect specific comorbidities and diseases, such as periodontitis. The limitation of available data of modern microbiomes may have limited the ability to compare results from individuals from Badia Pozzeveri to those of modern individuals. The present study will open the opportunity to augment the understanding of the lifestyle and health status of individuals from Badia Pozzeveri from the 11th to 19th centuries that would otherwise be difficult to obtain with archeological information alone.

## Figures and Tables

**Figure 1 genes-10-00299-f001:**
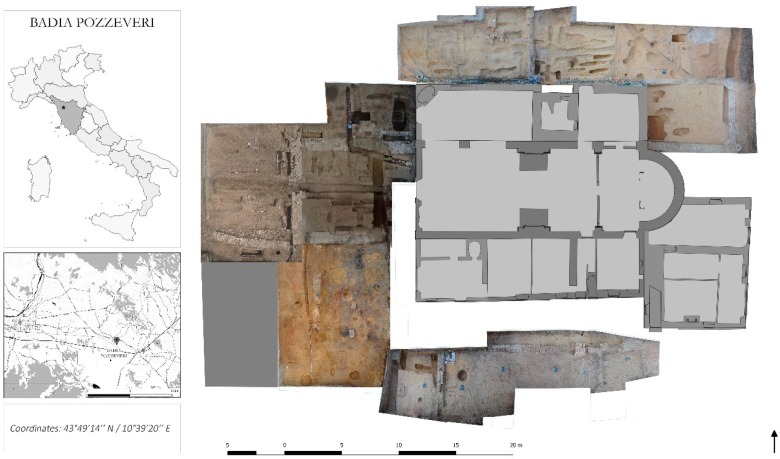
Geographical location and archeological site of San Pietro di Pozzeveri in the village of Badia Pozzeveri. On the left, geographical location of the archaeological site, which is located about 10 km southeast of the city of Lucca (Tuscany, Italy). On the right, representation of the archaeological excavation, which was conducted by the “open area method”.

**Figure 2 genes-10-00299-f002:**
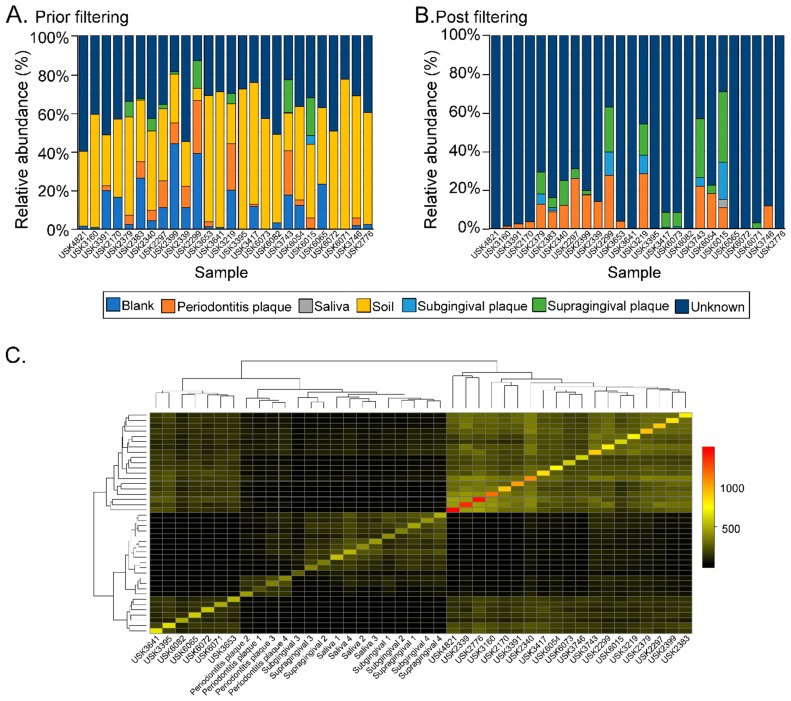
SourceTracker analyses. (Panel (**A**)) shows the relative abundance of the dental calculus OTUs resembling those of the extraction blank control, and dental plaque of subjects with periodontitis, saliva, soil, subgingival plaque, and supragingival plaque. Dental calculus operational taxonomic units (OTUs) that did not match the included microbiomes were classified as “unknown”. (Panel (**B**)) shows the relative abundance of OTUs in the dental calculus sample post-filtering OTUs identified in the soil samples from the archeological site, as well as the extraction blank control. (Panel (**C**)) shows a heatmap of the number of OTUs shared between the dental calculus and modern microbiome samples (i.e., saliva and dental plaque from healthy subjects and those with periodontitis).

**Figure 3 genes-10-00299-f003:**
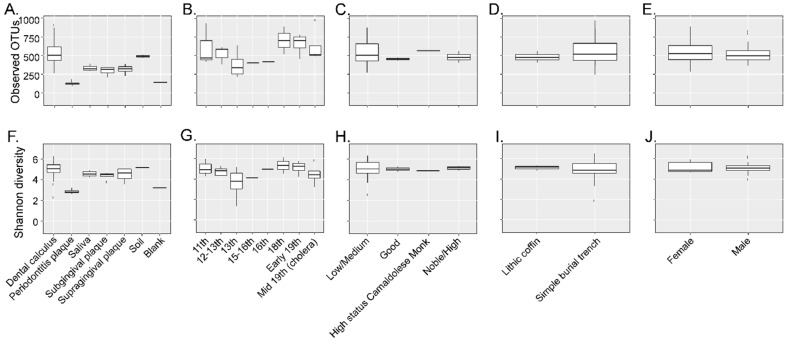
The α-diversity analyses. Observed OTUs were determined for the dental calculus samples based on sample type (Panel (**A**)), century (Panel (**B**)), social status (Panel (**C**)), burial type (Panel (**D**)), and gender (Panel (**E**)). Shannon diversity was determined based on sample type (Panel (**F**)), century (Panel (**G**)), social status (Panel (**H**)), burial type (Panel (**I**)), and gender (Panel (**J**)).

**Figure 4 genes-10-00299-f004:**
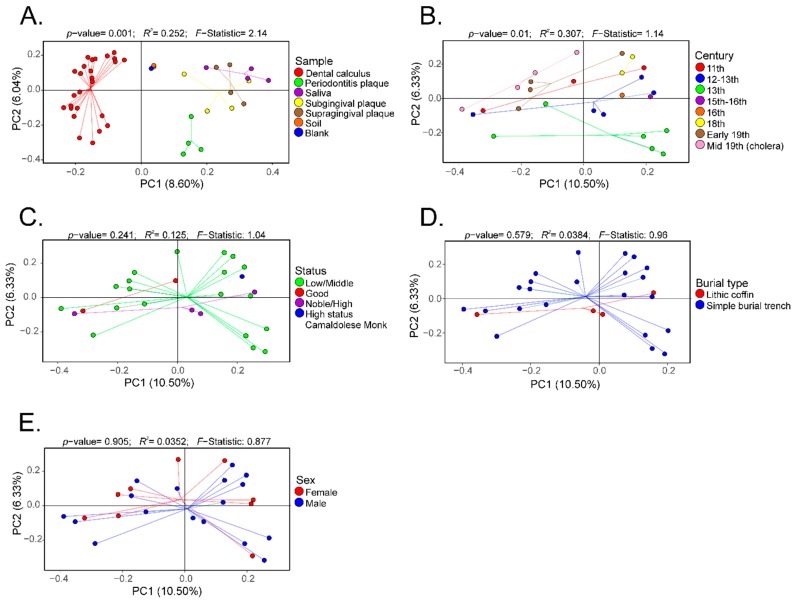
Principal coordinate analyses (PCoA) of the β-diversity (Bray–Curtis distances) of the dental calculus samples. The β-diversity analyses were performed based on sample type (Panel (**A**)), century (Panel (**B**)), social status (Panel (**C**)), burial type (Panel (**D**)), and gender (Panel (**E**)).

**Figure 5 genes-10-00299-f005:**
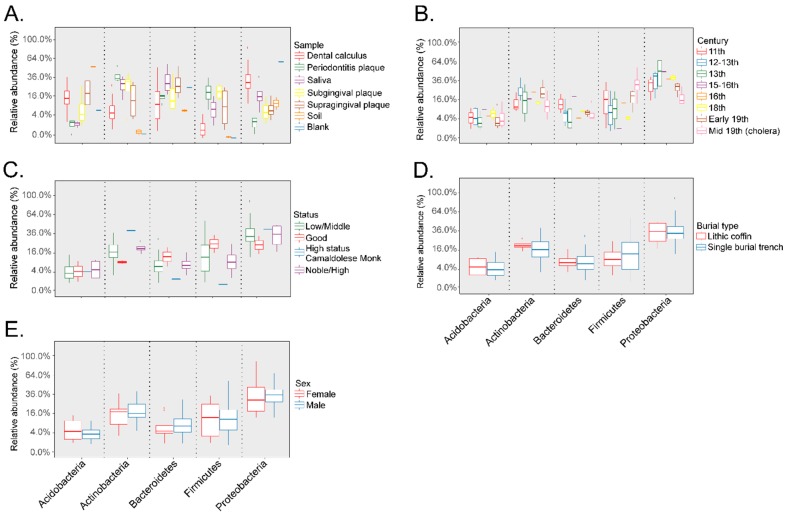
Relative abundances (%) of the top 5 main phyla in the dental calculus samples included Acidobacteria, Actinobacteria, Bacteroidetes, Firmicutes, and Proteobacteria. Taxonomy analyses were performed based on sample type (Panel (**A**)), century (Panel (**B**)), social status (Panel (**C**)), burial type (Panel (**D**)), and gender (Panel (**E**)).

**Table 1 genes-10-00299-t001:** Samples of dental calculus from Badia Pozzeveri. ID—identifier; F—female; M—male.

Sample ID	Sex	Age	Burial Typology	Chronology	Status
USK 3743	F	40–50	Simple burial trench	11th century	Good social status
USK 3746	M	30–40	Simple burial trench	11th century	Good social status
USK 2776	M	45–55	Simple burial trench	11th century	Middle/low status
USK 6073	M	25–30	Lithic coffin	12–13th century	Noble or high status
USK 4821	M	35–45	Simple burial trench	12–13th century	High-status Camaldolese monk
USK 3219	M	30–40	Lithic coffin	12–13th century	Noble or high status
USK 3395	F	30–40	Lithic coffin	12–13th century	Noble or high status
USK 3417	M	30–40	Lithic coffin	12–13th century	Noble or high status
USK 6082	M	20–25	Simple burial trench	13th century	Middle/low status
USK 6071	M	20–25	Simple burial trench	13th century	Middle/low status
USK 6054	M	35–45	Simple burial trench	13th century	Middle/low status
USK 6015	M	20–25	Simple burial trench	13th century	Middle/low status
USK 6065	F	40–50+	Simple burial trench	13th century	Middle/low status
USK 6072	M	40–50	Simple burial trench	13th century	Middle/low status
USK 3641	F	30–40	Simple burial trench	15th–16th century	Middle/low status
USK 3653	M	25–35	Simple burial trench	16th century	Middle/low status
USK 3391	M	30–40	Simple burial trench	18th century	Middle/low status
USK 3160	M	35–45	Simple burial trench	18th century	Middle/low status
USK 2299	M	25–30	Simple burial trench	1855 cholera	Middle/low status
USK 2339	F	60+	Simple burial trench	1855 cholera	Middle/low status
USK 2399	M	50–60	Simple burial trench	1855 cholera	Middle/low status
USK 2297	F	45–55	Simple burial trench	1855 cholera	Middle/low status
USK 2340	M	30–40	Simple burial trench	First half 19th century	Middle/low status
USK 2383	F	30–40	Simple burial trench	First half 19th century	Middle/low status
USK 2379	F	20–30	Simple burial trench	First half 19th century	Middle/low status
USK 2170	F	20–30	Simple burial trench	First half 19th century	Middle/low status

**Table 2 genes-10-00299-t002:** Group significance analyses based on sample type. Results show the average relative abundances of selected bacterial species that are shared between the dental calculus samples and the modern oral microbiomes including dental plaque from subjects with and without periodontitis, as well as saliva (shared species). Results also show selected bacteria that were only identified in the dental calculus samples (unique species). *p*-Value and False Discovery Rate (FDR)-Adjusted *p* values are shown.

	*p*-Value	FDR-Adjusted *p*	Dental Calculus	Periodontitis Plaque	Saliva	Subgingival Plaque	Supragingival Plaque
**Shared species**							
*Rothia aeria*	4.96 × 10^−1^	9.94 × 10^−1^	0.02	0.01	0.03	1.13	0.14
*Rothia dentocariosa*	4.15 × 10^−3^	2.05 × 10^−2^	0.21	0.00	0.27	2.66	9.09
*Corynebacterium durum*	7.79 × 10^−3^	3.57 × 10^−2^	0.15	0.01	0.05	0.42	3.14
*Bacteroides endodontalis*	4.50 × 10^−3^	2.18 × 10^−2^	0.01	2.35	0.18	4.83	0.14
*Solobacterium moorei*	3.77 × 10^−1^	9.23 × 10^−1^	0.01	0.04	0.02	0.00	0.00
*Rothia mucilaginosa*	9.37 × 10^−3^	4.21 × 10^−2^	0.03	0.01	0.27	0.26	0.13
*Capnocytophaga ochracea*	1.41 × 10^−3^	8.74 × 10^−3^	0.02	0.51	0.27	2.62	2.2
*Steptococcus oralis*	3.58 × 10^−1^	8.93 × 10^−1^	0.01	0.00	0.00	0.00	0.03
*Haemophilus parainfluenzae*	2.29 × 10^−5^	4.93 × 10^−4^	0.03	0.02	9.09	0.91	1.98
*Campylobacter rectus*	3.71 × 10^−2^	1.40 × 10^−1^	0.04	1.43	0.02	0.02	0.17
*Aggregatibacter segnis*	2.56 × 10^−1^	6.57 × 10^−1^	0.05	0.18	0.06	1.14	0.14
*Treponema socranskii*	1.70 × 10^−2^	7.01 × 10^−2^	0.52	2.40	0.04	0.45	0.52
*Neisseria subflava*	3.49 × 10^−2^	1.33 × 10^−1^	0.03	0.00	1.74	0.13	0.02
*Micrococcus luteus*	7.50 × 10^−1^	9.94 × 10^−1^	0.01	0.00	0.00	0.00	0.01
*Argemone mexicana*	8.06 × 10^−1^	9.94 × 10^−1^	0.03	0.00	0.00	0.01	0.00
*Desulfomicrobium orale*	5.74 × 10^−1^	9.94 × 10^−1^	0.06	0.01	0.00	0.00	0.00
*Kocuria palustris*	7.99 × 10^−1^	9.94 × 10^−1^	0.01	0.00	0.00	0.00	0.01
**Unique species**							
*Lentzea albidocapillata* ^1^	3.05 × 10^−2^	1.20 × 10^−1^	0.27	0.00	0.00	0.00	0.00
*Enterococcus cecorum* ^2^	9.60 × 10^−1^	9.94 × 10^−1^	0.05	0.00	0.00	0.00	0.00
*Enterococcus faecium* ^2^	9.94 × 10^−1^	9.94 × 10^−1^	0.01	0.00	0.00	0.00	0.00
*Burkholderia gladioli* ^3^	9.94 × 10^−1^	9.94 × 10^−1^	0.01	0.00	0.00	0.00	0.00
*Inquilinus limosus* ^3^	6.76 × 10^−1^	9.94 × 10^−1^	0.01	0.00	0.00	0.00	0.00
*Methylobacterium mesophilicum* ^1^	9.60 × 10^−1^	9.94 × 10^−1^	0.01	0.00	0.00	0.00	0.00
*Microbacterium oxydans* ^1^	1.56 × 10^−3^	8.94 × 10^−3^	0.17	0.00	0.00	0.00	0.00
*Tsukamurella pulmonis* ^3^	2.44 × 10^−1^	6.35 × 10^−1^	0.07	0.00	0.00	0.00	0.00
*Necropsobacter rosorum* ^1^	4.38 × 10^−1^	9.94 × 10^−1^	0.05	0.00	0.00	0.00	0.00
*Staphylococcus sciuri* ^1^	9.60 × 10^−1^	9.94 × 10^−1^	0.02	0.00	0.00	0.00	0.00
*Arthrobacter scleromae* ^1^	4.34 × 10^−1^	9.94 × 10^−1^	0.01	0.00	0.00	0.00	0.00

^1^ Opportunistic/nosocomial/other pathogens; ^2^ fecal bacteria; ^3^ respiratory-associated bacteria.

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
