# Peer review of "Commensal and Pathogenic Members of the Dental Calculus Microbiome of Badia Pozzeveri Individuals from the 11th to 19th Centuries"

_genes, 2019, doi:10.3390/genes10040299_

Round 1
Reviewer 1 Report
General comments.
I am grateful for having the opportunity to read this interesting paper. This microbiome-based study was very well-conducted, and the paper well-written.
The conclusions, mainly observational, provide new and precious data in the new field of anthropology that utilizes the ancient calculus microbiome as a new approach.
Major comments.
The paragraph lines 316-325 has to be toned down or enriched in terms of hypothesis.
The presence of DNA of opportunistic pathogens like B. gladioli does not suggest itself a specific thing. The number of explanations is so huge, that it sounds speculative to talk only about respiratory disorders. Indeed, the colonization rate of environmental pathogens (eg. Pseudomonas aeruginosa) in the oral or digestive tracts of healthy people is around 5% (intermittent colonization). One can imagine that these bacterial species might be found in dental samples too.
S. sciuri has also been described in animals, and does not have a specific tropism for the respiratory tract.
Minor comments.
1. Line 151: "using the Illumina MiSeq reagent kit V3 (2X150 bp) » ; It is is more likely 2x300 bp with the version 3.
2. Paragraph lines 316-325: bacterial names have to be in italics.
Author Response
Reviewer #1
Major comments.
The paragraph lines 316-325 has to be toned down or enriched in terms of hypothesis. The presence of DNA of opportunistic pathogens like B. gladioli does not suggest itself a specific thing. The number of explanations is so huge, that it sounds speculative to talk only about respiratory disorders. Indeed, the colonization rate of environmental pathogens (eg. Pseudomonas aeruginosa) in the oral or digestive tracts of healthy people is around 5% (intermittent colonization). One can imagine that these bacterial species might be found in dental samples too.
We thank the reviewer for these very helpful insights. We have clarified this throughout the discussion section and toned down the statements (Lines 349-354; 364-372).
S. sciuri has also been described in animals, and does not have a specific tropism for the respiratory tract.
We thank the reviewer for this statement. We have removed this statement for the modified version of the manuscript.
Minor comments.
1. Line 151: "using the Illumina MiSeq reagent kit V3 (2X150 bp) » ; It is is more likely 2x300 bp with the version 3.
We have modified accordingly (Line 176).
2. Paragraph lines 316-325: bacterial names have to be in italics.
We have modified this accordingly.
Reviewer 2 Report
Santiago-Rodriguez et al. characterize human dental calculus in 26 ancient individuals from medieval Italy alongside modern microbiome data to explore potential differences in time period, social status, sample type and sex in their data set. However, the differences in taxa present in the ancient calculus samples could not be explained due to any of the previously mentioned variables. The authors further explored the presence of specific commensal and pathogenic microbiome constituents within the individuals. While the work is potentially interesting, there needs to be a greater emphasis on conveying the methodological rigor and soundness of the ancient calculus results, as well as demonstrating the significance of the results to understanding medieval health at this site.
I have concerns about the authenticity of the results reported, particularly surrounding the implementation of protocols to minimize contamination and cross-contamination as well as downstream analyses that demonstrate true endogenous signals in the ancient calculus samples. The authors should provide further evidence that the findings are not due to contamination/cross-contamination.
Research design and protocols
o How was contamination and cross-contamination monitored throughout the experiment? This was not clearly indicated in the Methods, aside from an "extraction blank" (not indicated as whether it's from calculus or soil) and non-template controls (which were not included in downstream analyses).
o Were any teeth from these individuals sequenced (metagenomically), not just the calculus? If so, it would strengthen support that ancient calculus samples are not derived from contamination in the post-mortem environment.
Subsampling calculus
o P. 4, lines 116-117 - Recognizing the issue with sampling in as contained an environment as possible, could the authors elaborate on what was a “protected environment”?
o P. 4 line 119 - Removal of dental calculus, was this from supragingival or subgingival? Or a mixture? There are known differences between sub- and supra-gingival microbiota shown in modern studies, the authors should clarify their protocol
o P. 4 line 120 - Removing calculus from lingual or buccal surface – there is a potential association between tooth surface and microbiota as well as the type of tooth – please indicate the types of teeth and location for each from which calculus was removed. Relatedly, did the authors control for differences between tooth surface and microbiota present in the downstream analyses since there is a potentially significant association that may drive the diversity signal
o P. 4 line 125-126 – Could the authors clarify whether the calculus samples were decontaminated? If so, how?
Assessing contaminant DNA
o Generally, there are three types of negative controls minimally required to allow adequate monitoring of contaminants in microbiome work: 1) sampling blanks, 2) DNA extraction blanks, and 3) no-template amplification blank
o P. 4 line 134 indicates two soil samples and a blank control were extracted
o First, is this a blank control from the soil or from the calculus? What is its identifier in the manuscript? It cannot be discerned from the figures (e.g., Fig. 2)
o Second, there is a ratio in which to include extraction controls (i.e., one extraction control for every 8 samples for example) – why was only one extraction blank used to monitor contamination through the whole experiment? There is also mention of non-template PCR controls (P. 4 line 141) – these do not appear to have been analyzed.
o Third, were the soil samples processed at the same time as the calculus samples? The impact of cross-contamination from soil, which is of a heavier biomass, is significant.
o The authors should address how their study meets expected standards for monitoring contamination/cross-contamination from sampling to downstream analyses, particularly whether the control data (one extraction blank) is sufficient for this.
Determining the level of contamination
o Could the authors assess if the OTUs present in the blank control impacted the interpretation of OTUs in the ancient calculus samples? The pre- and post-filtering counts do not convey how the level of contamination was established so as to avoid including samples that were contaminated or had low endogenous signals
o Relatedly, the authors should provide more detail on how the contaminant taxa were explored to demonstrate the impact on the interpretation of results (aside from comparing the five most common phyla)
o It would be useful to include the taxonomic profile of the blank itself, beyond the five most common phyla, to ensure differences in taxa abundance/composition are not driven by contaminant DNA
o P. 5 line 163 – in using SourceTracker to establish the contaminant profile, can the authors comment on how well contaminant DNA sequences were removed? Is it that these different communities from the different sources (i.e., soil, modern microbiomes, blank) acted to mimic/parallel the actual microbial community for the ancient samples so that contaminants could be removed? Greater detail here would be appreciated, particularly as cross-contamination can confound this analysis.
Results
o How distinct were the ancient calculus results from laboratory and environment contamination (e.g., blank, soil)? This would help support statements regarding the distinguishability of ancient biological signals from soil and laboratory contaminants
o Although the ancient and modern calculus samples share OTUs, 19 of the ancient samples had higher proportion of OTUs and clustered separately from the modern samples
o If the authors are suggesting a ubiquity of oral microbiome components (but differences in relative abundance), this result/statement seems contrary. Could the authors provide further detail to elaborate.
o How have the authors established that these signals are authentic? Some are within a genus associated with known contaminant DNA in the lab environment (e.g., Neisseria, Enterococcus, Veillonela, Lactobaccilus)
o The proportion of OTUs assigned to the commensal and pathogenic microbial constituents should be identified – not just a presence absence, but proportion of overall taxa identified
o Burkholderia gladioli, Methylobacterium mesophilicum are also found in soil – how can the authors be certain that the signal is not from soil and it is indeed human-associated?
o How do the pathogens identified fit into the context for what is known about Badia Pozzeveri? For example, line 304-305 (P. 11) about hygiene practices seems incomplete, just on the basis of Enterococcus bacteria. How many individuals were the suite of pathogens found in? The context for the importance of these results should be established, since one of the aims of the manuscript is to enhance the understanding of lifestyle and health from the individuals associated with this site.
o Exploring the potential changes in the oral microbiome by looking at socioeconomic status, dietary habits etc. was limited in scope, but it would be helpful to see how the bacterial taxa present changed in time (i.e., abundance/proportion) even if there was no association with the studied variables.
Minor comments
o P. 4, line 117 – What is meant by “cuff”? Does this mean sterile sleeves were worn?
o P. 5 lines 156-158 - Regarding the inclusion of modern microbiome data – the authors should provide additional information on the context of these samples (i.e., geographical location, type of sequencing/library prep, type of sample) in the supplementary.
o P. 7 line 230 – Figure 3 the p-value is 0.01, not 0.001
o P.10 – In Table 2, what are the units of the values for each of the sample types?
o Page 11, line 307 - Is there osteological data for these individuals that can be integrated on the findings of diseases, particularly periodontitis?
Author Response
Reviewer #2
Research design and protocols
o How was contamination and cross-contamination monitored throughout the experiment? This was not clearly indicated in the Methods, aside from an "extraction blank" (not indicated as whether it's from calculus or soil) and non-template controls (which were not included in downstream analyses).
Prior monitoring contamination and cross-contamination using PCR and sequencing, previously suggested guidelines were applied. For instance, the use of gloves for the collection of each sample, sterilization of scalpels, the use of individualized collection tubes, etc. Indeed, contamination and cross-contamination was monitored by sequencing the extraction blank and the addition of a non-template PCR reaction. DNA from the dental calculus samples were extracted using the DNA Investigator Kit, and the DNA from the soil samples were extracted using the Power Soil kit. We have included this in the revised version of the manuscript. Non-template controls were not sequenced because of a lack of amplification and lack of a DNA signal (Lines 153-157; 164-166).
o Were any teeth from these individuals sequenced (metagenomically), not just the calculus? If so, it would strengthen support that ancient calculus samples are not derived from contamination in the post-mortem environment.
We thank the reviewer for this important suggestion. We understand that it has been suggested that shotgun metagenomic sequencing is a more powerful tool than 16S rRNA amplicon sequencing for microbiome studies mostly because it can overcome the biases introduced during the PCR amplification step. However, shotgun metagenomic sequencing may have not necessarily augment the evidence that these samples have suffered post-portem contamination. In fact, 16S rRNA gene high-throughput sequencing may perhaps be a better tool to demonstrate this because whole-genome sequence databases are still lagging in the number of complete genomes sequenced compared to the number of available 16S rRNA gene sequences in databases. Some of the OTUs present in the dental calculus samples even after filtering OTUs present in the soil sample from the archeological site is evidence that these samples have been impacted by post-portem contamination. However, we have focused on those OTUs that are part of the human oral and nasopharynx microbiome. We re-ensured to include this in the revised version of the manuscript (Lines 317-322).
Subsampling calculus
o P. 4, lines 116-117 - Recognizing the issue with sampling in as contained an environment as possible, could the authors elaborate on what was a “protected environment”?
A protected environment refers to the use of a vertical laminar flow safety hood. We have clarified this in the revised version of the manuscript (Line 131).
o P. 4 line 119 - Removal of dental calculus, was this from supragingival or subgingival? Or a mixture? There are known differences between sub- and supra-gingival microbiota shown in modern studies, the authors should clarify their protocol
We have clarified that the calculus samples consisted of supragingival plaque from molars and pre-molars (Lines 134-135).
o P. 4 line 120 - Removing calculus from lingual or buccal surface – there is a potential association between tooth surface and microbiota as well as the type of tooth – please indicate the types of teeth and location for each from which calculus was removed. Relatedly, did the authors control for differences between tooth surface and microbiota present in the downstream analyses since there is a potentially significant association that may drive the diversity signal
We thank the reviewer for this insight. We have clarified in the revised version of the manuscript that the dental calculus samples were removed from the lingual side of the mandibular molars and premolars (Lines 135-136).
o P. 4 line 125-126 – Could the authors clarify whether the calculus samples were decontaminated? If so, how?
Calculus samples were decontaminated prior DNA extraction using UV radiation for 1 min (Lines 147-148).
Assessing contaminant DNA
o Generally, there are three types of negative controls minimally required to allow adequate monitoring of contaminants in microbiome work: 1) sampling blanks, 2) DNA extraction blanks, and 3) no-template amplification blank
We included a DNA extraction blank, as well as a no-template amplification blank as recommended in (Weyrich et al. 2018). We included the DNA extraction blank for sequencing, and a non-template PCR reaction, which showed no band and no DNA signal when measuring the DNA. Since DNA in the no-template amplification blank showed no detectable signal when measuring the DNA fluorometrically, it was not included in the sequencing pool. We have clarified this in the revised version of the manuscript (Lines 153-157; 164-166).
Weyrich, L., Farrer, A. G., Eisenhofer, R., Arriola, L. A., Young, J., Selway, C. A., ... & Cooper, A. (2018). Laboratory contamination over time during low-biomass sample analysis. BioRxiv, 460212.
o P. 4 line 134 indicates two soil samples and a blank control were extracted
Indeed, two soil samples were sequenced. In the process of analyzing the data, the mapping file used for the QIIME analyses categorize both samples as “Soil”. In categorizing both soil samples as “Soil”, these were merged during the alpha-diversity and taxonomic analysis. In the PCoA plot of the beta-diversity, however, two sample points are shown for both soil samples (Figure 4A).
o First, is this a blank control from the soil or from the calculus? What is its identifier in the manuscript? It cannot be discerned from the figures (e.g., Fig. 2)
We have clarified in the revised version of the manuscript that the blank was from the calculus samples (Lines 153-157).
o Second, there is a ratio in which to include extraction controls (i.e., one extraction control for every 8 samples for example) – why was only one extraction blank used to monitor contamination through the whole experiment? There is also mention of non-template PCR controls (P. 4 line 141) – these do not appear to have been analyzed.
Indeed, blank controls were included for every 2-3 samples, but due to budget limitations, we were only able to sequence one blank control. In terms of the no-template PCR control, it was included in the PCR reaction, and products were quantified, but no detectable DNA signal was produced. We have included this in the revised version of the manuscript (Lines 164-166).
o Third, were the soil samples processed at the same time as the calculus samples? The impact of cross-contamination from soil, which is of a heavier biomass, is significant.
The soil sample was processed separately from the dental calculus samples. We have clarified this in the revised version of the manuscript (Lines 154-155).
o The authors should address how their study meets expected standards for monitoring contamination/cross-contamination from sampling to downstream analyses, particularly whether the control data (one extraction blank) is sufficient for this.
We understand the reviewer’s concern. It has been suggested that at least two controls should be included in ancient microbiome studies (Weyrich et al., 2018): the extraction blank control and a non-template PCR reaction. Both were included in the study. The blank was sequenced, and the non-template PCR reaction was not sequenced because it did not show a band or a DNA signal (Lines 153-157; 164-166; 301-302).
Weyrich, L., Farrer, A. G., Eisenhofer, R., Arriola, L. A., Young, J., Selway, C. A., ... & Cooper, A. (2018). Laboratory contamination over time during low-biomass sample analysis. BioRxiv, 460212.
Determining the level of contamination
o Could the authors assess if the OTUs present in the blank control impacted the interpretation of OTUs in the ancient calculus samples? The pre- and post-filtering counts do not convey how the level of contamination was established so as to avoid including samples that were contaminated or had low endogenous signals
We thank the reviewer for pointing this. We have performed additional analyses to compare the similarity across samples by plotting beta-diversity values in a PCoA plot. Analysis was performed with the dental calculus samples prior and after filtering the taxa identified in the blank control. In new Supplementary Figure 1, PCoA of the beta-diversity (Panel A) was performed to show the similarity between the dental calculus samples and the blank control. Permanova results show that the dental calculus and blank control are significantly different, suggesting that the blank control did not significantly affect the interpretation of the results.
In addition, PCoA of the beta-diversity prior and after filtering the blank and soil samples showed that the dental calculus samples prior filtering the blank and soil samples cluster very closely to the mentioned controls. After filtering the OTUs identified in the blank and soil samples, the dental calculus samples clustered separately from the samples prior filtering and from the blank and soil samples (Supplementary Figure 1B). When plotting the percentage of OTUs prior and after filtering, as well as those identified in the blank control and soil samples, results show that soil may have had a greater number of identified OTUs impacting the dental calculus samples (Supplementary Figure 1C and New Dataset 4).
We have included this in the revised version of the manuscript (Lines 231-243).
o Relatedly, the authors should provide more detail on how the contaminant taxa were explored to demonstrate the impact on the interpretation of results (aside from comparing the five most common phyla)
We thank the reviewer for this suggestion. We have now performed additional analyses and have included them as part of new Supplementary Figure 1 and Dataset 4.
o It would be useful to include the taxonomic profile of the blank itself, beyond the five most common phyla, to ensure differences in taxa abundance/composition are not driven by contaminant DNA
We agree with the reviewer that the taxonomic profile of the blank control needs to be included in the manuscript. We have included the taxonomic profile of the extraction blank control and can be found in Dataset 1.
o P. 5 line 163 – in using SourceTracker to establish the contaminant profile, can the authors comment on how well contaminant DNA sequences were removed? Is it that these different communities from the different sources (i.e., soil, modern microbiomes, blank) acted to mimic/parallel the actual microbial community for the ancient samples so that contaminants could be removed? Greater detail here would be appreciated, particularly as cross-contamination can confound this analysis
The SourceTracker analyses are a way to estimate what percentage of OTUs in a ‘source’ are present in a sample of interest. SourceTracker analyses are useful in ancient microbiome analyses because it estimates the percentage of OTUs of a sample specified as a ‘source’ (i.e. blank and soil) are present in a sample of interest (i.e. dental calculus). The SourceTracker script itself does not remove the OTUs present in the ‘source’ from the sample of interest; rather, it would be the series of script described in http://qiime.org/tutorials/filtering_contamination_otus.html. We have clarified this in the revised version of the manuscript (Line 193; 197-198).
Results
o How distinct were the ancient calculus results from laboratory and environment contamination (e.g., blank, soil)? This would help support statements regarding the distinguishability of ancient biological signals from soil and laboratory contaminants
Please see response above.
o Although the ancient and modern calculus samples share OTUs, 19 of the ancient samples had higher proportion of OTUs and clustered separately from the modern samples
We thank the reviewer for pointing this. This has been included in the manuscript (Lines 245-248).
o If the authors are suggesting a ubiquity of oral microbiome components (but differences in relative abundance), this result/statement seems contrary. Could the authors provide further detail to elaborate?
We thank the reviewer for pointing this. We agree that the mentioned factors can affect the relative abundances of some of these microbes. We refer more to the presence of these microbes rather than relative abundances. We have clarified this in the revised version of the manuscript (Line 331-332).
o How have the authors established that these signals are authentic? Some are within a genus associated with known contaminant DNA in the lab environment (e.g., Neisseria, Enterococcus, Veillonela, Lactobacillus)
We thank the reviewer for pointing this. We have based that these genera are part of the human gut, oral and nasopharynx microbiomes based on previous studies characterizing these microbiomes in modern or extant individuals. We have included references supporting the presence of these genera as part of the human microbiome in the revised version of the manuscript (Lines 280-292).
o The proportion of OTUs assigned to the commensal and pathogenic microbial constituents should be identified – not just a presence absence, but proportion of overall taxa identified
We thank the reviewer for pointing this. The average relative abundances of the selected commensal and pathogenic OTUs identified in the dental calculus samples are shown in Table 2, but it was not clear what the numbers meant. We have clarified this in the revised version of the manuscript (Line 280).
o Burkholderia gladioli, Methylobacterium mesophilicum are also found in soil – how can the authors be certain that the signal is not from soil and it is indeed human-associated?
We agree with the reviewer. Although these species have been identified as opportunistic pathogens, they may also inhabit other environments. We have toned down the statement to specify in the revised version of the manuscript that some of these microbes can be present in environments outside the human oral cavity and nasopharynx (Line 349-354; 364-372).
o How do the pathogens identified fit into the context for what is known about Badia Pozzeveri? For example, line 304-305 (P. 11) about hygiene practices seems incomplete, just on the basis of Enterococcus bacteria. How many individuals were the suite of pathogens found in?
The context for the importance of these results should be established, since one of the aims of the manuscript is to enhance the understanding of lifestyle and health from the individuals associated with this site.
We thank the reviewer for pointing this. The sampled individuals did not show dental abscesses, inflammatory signs (cribrosity) of alveolitis or periodontitis. We have included this in the revised version of the manuscript (Lines 132-133; 352-354).
o Exploring the potential changes in the oral microbiome by looking at socioeconomic status, dietary habits etc. was limited in scope, but it would be helpful to see how the bacterial taxa present changed in time (i.e., abundance/proportion) even if there was no association with the studied variables.
We agree with the reviewer in that time is an interesting variable to be considered. We are interested in the microbiome and specific bacterial taxa changes throughout time. For this reason, we included the relative proportions based on century as part of Dataset 7. Because most of the bacterial taxa randomly showed at specific centuries and not others, made it difficult to explain the results and we decided not to expand on this point on the manuscript.
Minor comments
o P. 4, line 117 – What is meant by “cuff”? Does this mean sterile sleeves were worn?
We thank the reviewer for pointing this. We have clarified that surgical gowns were used (Line 132).
o P. 5 lines 156-158 - Regarding the inclusion of modern microbiome data – the authors should provide additional information on the context of these samples (i.e., geographical location, type of sequencing/library prep, type of sample) in the supplementary.
We thank the reviewer for these suggestions. We have added Supplementary Table 2 with further information about the public sequence data used for comparisons with the dental calculus samples (Lines 188-189).
o P. 7 line 230 – Figure 3 the p-value is 0.01, not 0.001
We have modified accordingly.
o P.10 – In Table 2, what are the units of the values for each of the sample types?
We have clarified that these are the average of the relative abundances.
o Page 11, line 307 - Is there osteological data for these individuals that can be integrated on the findings of diseases, particularly periodontitis?
We thank the reviewer for pointing this. Since this is an ongoing excavation, we are still in the process of gathering the data; thus, will not be included in the manuscript at the moment.
Round 2
Reviewer 2 Report
I very much appreciate the attention to the comments, particularly conducting additional analyses and providing clarification where needed. The availability of the extra supplementary data files are indeed helpful and strengthen the scope of the analysis.
I have a few minor comments:
P. 4, lines 139-141 - It may be worth mentioning that 2-3 blanks were included per sample, but only one was sequenced.
P. 6, line 217 - When referring to Supplementary Figure 1B, it would be helpful to provide the p-value, F-statistics etc., similar to the information provided for Supplementary Figure 1A on line 212.
P. 12 - Italicize pathogen names where identified
It would be helpful to directly state somewhere in the results or discussion that the OTUs focused on are those specific (or unique to) the human oral and nasopharynx microbiome, just to clarify the analyses.
Author Response
We thank the reviewer for additional suggestions to improve the manuscript.
P. 4, lines 139-141 - It may be worth mentioning that 2-3 blanks were included per sample, but only one was sequenced.
We thank the reviewer for this suggestion. We have included this information in the revised version of the manuscript (Line 139-140).
P. 6, line 217 - When referring to Supplementary Figure 1B, it would be helpful to provide the p-value, F-statistics etc., similar to the information provided for Supplementary Figure 1A on line 212.
We have included this information in the revised version of the manuscript (Lines 218-219).
P. 12 - Italicize pathogen names where identified
We have modified accordingly.
It would be helpful to directly state somewhere in the results or discussion that the OTUs focused on are those specific (or unique to) the human oral and nasopharynx microbiome, just to clarify the analyses.
We thank the reviewer for this insight. We have directly stated this on the Results and Discussion section. (Lines 279-280;329).